# Leveraging Advances in Artificial Intelligence to Improve the Quality and Timing of Palliative Care

**DOI:** 10.3390/cancers12051149

**Published:** 2020-05-03

**Authors:** Paul Windisch, Caroline Hertler, David Blum, Daniel Zwahlen, Robert Förster

**Affiliations:** 1Department of Radiation Oncology, Kantonsspital Winterthur, 8400 Winterthur, Switzerland; daniel.zwahlen@ksw.ch (D.Z.); robert.foerster@ksw.ch (R.F.); 2Competence Center for Palliative Care, University Hospital Zurich, 8091 Zurich, Switzerland; Caroline.Hertler@usz.ch (C.H.); david.blum@usz.ch (D.B.)

In recent years, research on artificial intelligence (AI) in medicine has seen great advances, especially with regards to the detection of diseases [1,2]. While definitions of what exactly constitutes AI vary, most definitions mention computer-based systems solving tasks that would normally require “natural”, especially human, intelligence [3]. Among the many subspecialities of AI, deep learning, where information from the input is extracted through many layers normally by so called neural networks, has emerged as an important tool due to its ability to extract meaningful information from imaging data [4].

State-of-the-art deep learning algorithms have been able to compete with—and in some cases even surpass—trained physicians in terms of diagnostic accuracy for certain indications [1,2].

While other areas of interest such as drug discovery have emerged, the application of AI is still largely limited to projects with the potential for great commercial gain. In contrast, research on its impact in other fields such as global health is comparatively slow [5,6].

One of these fields that has not been the focus and therefore has been largely unaffected by the recent advances is palliative care, a discipline of increasing importance in the aging population of the industrialized nations [7]. Palliative care is an interdisciplinary concept that strives to improve quality of life and prevent or alleviate symptoms for patients with severe, complex, and in some cases terminal illnesses [8].

As of March 2020, searching PubMed for publications whose titles contain the word “palliative” and either of the terms “deep learning”, “machine learning” or “artificial intelligence” yielded only four publications [9,10,11,12]. One of these publications was a rapid review searching databases for publications that used machine learning to improve palliative care that found only three publications trying to predict short-term mortality [10].

However, as palliative care is becoming more and more relevant, the possibilities to apply AI techniques to the research questions of the field increase drastically. Since many AI methods require massive amounts of learning data to reach their full predictive potential, well-curated datasets are a mandatory prerequisite [4]. As “conventional”, non-AI research in palliative care becomes increasingly established due to more patients that can be analyzed as well as more funding and interest, those datasets that are created as a byproduct can be used as a starting point for introducing AI to the field. If initial results are promising, a collaboration of palliative care centers to create even larger datasets could further improve these results. Especially imaging studies using deep learning could benefit from more generalizable predictions if the training images are acquired on a variety of heterogeneous scanners instead of the same machines from a single hospital [4]. To reduce the amount of data needed for the initial studies, researchers should also take advantage of advances in AI research such as image augmentation that apply modifications to training images to expand those datasets [13].

One challenge of particular interest could be the determination of the optimal timing of palliative care involvement. A growing number of publications have confirmed the positive impact of early palliative care for patients with cancer [14]. Likewise, the American Society of Clinical Oncology (ASCO) has implemented the recommendation for the integration of palliative care into standard oncology care beyond end-of-life care in 2012 and updated this recommendation to further strengthen the role of palliative care and detail what it should consist of in 2016 [15,16].

However, the recommended involvement of palliative care in the early course of the disease is often not feasible as the capacity of many palliative care facilities or the availability is still limited. Additionally, patients may be reluctant to receive palliative care, especially when the symptom burden is still low.

It is therefore of great importance to adequately time the involvement of palliative care well, allowing one to allocate resources efficiently while simultaneously avoiding any compromise to the patients’ quality of life.

Current approaches to this problem are largely based on clinical scores that try to predict mortality [17,18].

While the timing of palliative care is certainly an important aspect that could benefit from AI, it is far from being the only one. Every decision where the clinician has to weigh the possible benefits of an intervention with the stress caused by performing it could benefit from better predictions of whether the benefits that doctor and patient hope for will actually occur. This includes predicting if the pain caused by a bone metastasis will respond to palliative radiation or how long whole-brain radiotherapy can delay further neurologic deterioration. One might also try to assess more accurately whether palliative chemotherapy will result in a quicker decline in quality of life than the disease itself for a given patient.

Using AI to address these problems could help improve the current predictions by not only incorporating clinical but also imaging data, thereby not only predicting mortality but also the probability of increased symptom burden and a decrease in quality of life throughout the course of a patient’s disease. 

This use of AI for palliative care is especially promising in a time where advances in model explainability such as Grad-CAM, which displays the areas of an image that a neural network bases its predictions on, and bayesian neural networks, which quantify the certainty of a model in its predictions, can be implemented much more easily than before, thereby helping physicians to not just blindly trust the artificial intelligence’s decisions but also to assess it and to ultimately incorporate its advice into a clinical decision of when to suggest which modality of palliative care to a patient [19,20].

As detecting diseases is already a major focus of AI research in medicine and the suggested use cases for leveraging AI in palliative care could impact adequate symptom management as well as advanced care planning and end of life care, one can hope for improvements in all areas of palliative care using the different modalities of AI.

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
