# Peer review of "Leveraging Advances in Artificial Intelligence to Improve the Quality and Timing of Palliative Care"

_cancers, 2020, doi:10.3390/cancers12051149_

Round 1

Reviewer 1 Report

This commentary focuses on artificial intelligence (AI) and palliative care (PC). The following points will help strengthen the commentary.

1) Paragraph 4: Could authors ensure that information about the state of literature on AI and PC is accurately conveyed? Even one of the references they cite (Storick et al. 2019) is a rapid review that finds 3 studies in the area.
2) Paragraph 5: Please provide a definition of PC. What are essential components of PC and which of those elements would benefit most from AI?
3) Paragraph 6: As before, is optimal timing of PC the only or most important element that could benefit from AI? Why? Are there other elements?
4) Paragraph 6: What about the ASCO update?
5) Paragraphs 7-11: Could authors be more explicit in terms of the literature about AI and the prediction of mortality risk, for example, over here, in general, and in terms of cancer patients, in particular, and tie this back to the main motivation for their commentary?

Author Response

Dear reviewer,

thank you for your in-depth review of our commentary and your suggestions to strengthen it that are highly appreciated and have caused us to make the following changes and additions: 

1) Paragraph 4: Could authors ensure that information about the state of literature on AI and PC is accurately conveyed? Even one of the references they cite (Storick et al. 2019) is a rapid review that finds 3 studies in the area

Response: We have added a sentence summarizing this study: 

“One of those publications was a rapid review searching databases for publications that used machine learning to improve palliative care that found only three publications trying to predict short-term mortality [10].”

We think that this result and given the comprehensive search that Storick et al. conducted and the amount of papers they screened supports the notion that palliative care is an underserved area when it comes to artificial intelligence. 

2) Paragraph 5: Please provide a definition of PC. What are essential components of PC and which of those elements would benefit most from AI?

Response: We have added a definition:

“Palliative care is an interdisciplinary concept that strives to improve quality of life and prevent or alleviate symptoms for patients with severe, complex and in some cases terminal illnesses [8].”

For the essential components that could benefit from AI please see 3).

3) Paragraph 6: As before, is optimal timing of PC the only or most important element that could benefit from AI? Why? Are there other elements?

Response: This is a very valuable remark which is why we have added other suggestions for possible applications of AI:

“While the timing of palliative care is certainly an important aspect that could benefit from AI, it is far from being the only one. Every decision where the clinician has to weigh the possible benefits of an intervention with the stress caused by performing it could benefit from better predictions of whether the benefits that doctor and patient hope for will actually occur. This includes predicting if the pain caused by a bone metastasis will respond to palliative radiation or how long whole brain radiotherapy can delay further neurologic deterioration. One might also try to assess more accurately whether palliative chemotherapy will result in a quicker decline in quality of life than the disease itself for a given patient.”

4) Paragraph 6: What about the ASCO update?

Response: We have updated the respective sentence:

“Likewise, the American Society of Clinical Oncology (ASCO) has implemented the recommendation for integration of palliative care into standard oncology care beyond end-of-life care in 2012 and updated this recommendation to further strengthen the role of palliative care and detail what it should consist of in 2016 [14,15]. ”

5) Paragraphs 7-11: Could authors be more explicit in terms of the literature about AI and the prediction of mortality risk, for example, over here, in general, and in terms of cancer patients, in particular, and tie this back to the main motivation for their commentary?

Response: We have added a statement on the fact that most AI research in palliative care at the moment is actually focussed on predicting (short-term) mortality:

“One of those publications was a rapid review searching databases for publications that used machine learning to improve palliative care that found only three publications trying to predict short-term mortality [10].”

While we of course welcome the fact that AI is used in palliative care research, we tried to emphasize the in our opinion even more underserved aspects such as predicting increased symptom burden (i.e. clinical deterioration) or predicting the benefit of interventions that are often performed for palliative patients. 

Thank you for your contributions.

Reviewer 2 Report

There is a lack of supporting literature and arguments in the article. Also the proposed method is not stated clearly.   

At first, a more comprehensive literature review is required on the application of AI in health science. AI is becoming popular in medical research, including many applications in different fields including imaging recognition, disease detection, personalized medicine, etc.         

Second, it is not clear about the current research on palliative care using AI. They only pointed out the number of related publications. However, they did not focus on the research itself and the future research direction in this area. 

Third, AI is not introduced and/or clearly stated. They need to provide the reader with knowledge about the basics of AI and the merits of AI in data analysis.  

Author Response

Dear reviewer,

thank you for your in-depth review of our commentary and your suggestions to elaborate on definitions and the underlying literature that are highly appreciated and have caused us to make the following changes and additions: 

1) There is a lack of supporting literature and arguments in the article. Also the proposed method is not stated clearly.   

Response: We have added supporting literature while making additions to the points you raised below (see 2-4). Stating the proposed method more clearly is difficult since the exact modality of AI that one should leverage depends on the research question one is trying to answer. For example, deep learning works well with images and lots of data points while more traditional machine learning algorithms such as random forests tend to produce good results with structured tabular data. We have however included more use cases where AI could be considered to improve palliative care research:

“While the timing of palliative care is certainly an important aspect that could benefit from AI, it is far from being the only one. Every decision where the clinician has to weigh the possible benefits of an intervention with the stress caused by performing it could benefit from better predictions of whether the benefits that doctor and patient hope for will actually occur. This includes predicting if the pain caused by a bone metastasis will respond to palliative radiation or how long whole brain radiotherapy can delay further neurologic deterioration. One might also try to assess more accurately whether palliative chemotherapy will result in a quicker decline in quality of life than the disease itself for a given patient.”

2) At first, a more comprehensive literature review is required on the application of AI in health science. AI is becoming popular in medical research, including many applications in different fields including imaging recognition, disease detection, personalized medicine, etc.

Response: We agree that AI is becoming increasingly popular in medical research. The decision to focus on image recognition and disease detection in our introducing paragraphs are, however, deliberate as the largest amount of impressive results has been obtained in those areas. While personalized medicine is certainly an area where AI should be able to strive, convincing evidence that it is actually able to significantly influence outcomes, in a way that classical statistics is unable to do, is still pending. We would therefore prefer to convince our readers to consider AI for palliative care research by focusing on an area where its impact has been established instead of other areas where hype and actualy merit have yet to be fully assessed.          

3) Second, it is not clear about the current research on palliative care using AI. They only pointed out the number of related publications. However, they did not focus on the research itself and the future research direction in this area. 

Response: We have included the contents of a review on that topic to provide additional information on the nature of the research that is being conducted:

“One of those publications was a rapid review searching databases for publications that used machine learning to improve palliative care that found only three publications trying to predict short-term mortality [8].” 

Assessing the future research direction of a field where so little work is being published remains difficult. The small amount of research being published was the exact reason why we wrote this comment with the goal of steering more research in this direction. 

4) Third, AI is not introduced and/or clearly stated. They need to provide the reader with knowledge about the basics of AI and the merits of AI in data analysis.  

Response: We have added a definition of AI and why deep learning in particular has gained so much traction in clinical medicine:

“While definitions of what exactly constitutes AI vary, most definitions mention computer-based systems solving tasks that would normally require ‘natural’, especially human, intelligence [3]. Among the many subspecialities of AI, deep learning, where information from the input is extracted through many layers normally by so called neural networks, has emerged as an important tool due to its ability to extract meaningful information from imaging data [4].”

The merits are illustrated by  two of the most influential deep learning papers in medicine that have been published to date:

“State-of-the-art deep learning algorithms have been able to compete with - and in some cases even surpass trained physicians in terms of diagnostic accuracy for certain indications [1,2].”

Thank you for your contributions.

Reviewer 3 Report

This is a very interesting topic and I congratulate the authors for driving the palliative care professionals and researcher’s attention for this tremendously important issue.

Line 2

« …algorithms have been been…»

(delete one ‘been’)

Line 24:

« …availability isd still limited. » 

(substitute is instead of ‘isd’)

Lines 25-26:

«This is underlined by the estimation that less than half of the hospital admission requiring palliative care actually receive it [6]. »

The precise reference is the original one: «However, data from the National Palliative Care Registry estimates that, despite increasing access, less than half of the 7-8% of all hospital admissions that need palliative care actually receive it [3]. »

However, this reference [3] is the palliative care, report card 2015 and the 2019 report is already available. Please take this fact into consideration.

Reading your paper was enthusiastic.

In paragraph number 10, I was wondering if «personal resources associated with improved QoL near end-of-life may include religiosity and beliefs, ‘‘acceptance of reality’’, ‘‘life meaning and purpose’’, ‘‘self-worth’’, ‘‘hope’,’ and ‘‘caregivers’ support and acceptance’’» [ (Storick et al, 2019), page 9], could also, or not, be included in palliative care artificial intelligence studies, although health records seldom have that kind of information.

Thank you.  

Author Response

Dear reviewer, 

thank you for your in-depth review of our commentary as well as the positive assessment of our work and its purpose. Your suggestions to further strengthen it are highly appreciated and have caused the following changes and additions:

1) Line 2

« …algorithms have been been…»

(delete one ‘been’)

Response: Done as suggested.

2) Line 24:

« …availability isd still limited. »

(substitute is instead of ‘isd’)

Response: Done as suggested.

3) Lines 25-26:

«This is underlined by the estimation that less than half of the hospital admission requiring palliative care actually receive it [6]. »

The precise reference is the original one: «However, data from the National Palliative Care Registry estimates that, despite increasing access, less than half of the 7-8% of all hospital admissions that need palliative care actually receive it [3]. »

However, this reference [3] is the palliative care, report card 2015 and the 2019 report is already available. Please take this fact into consideration.

Respose: As the most recent report card does not contain a similar statement, we have removed that sentence from our commentary. Instead, we have added the 2019 report as a reference to the previous sentence as it underlines that “the capacity of many palliative care facilities or the availability is still limited.”

4) In paragraph number 10, I was wondering if «personal resources associated with improved QoL near end-of-life may include religiosity and beliefs, ‘‘acceptance of reality’’, ‘‘life meaning and purpose’’, ‘‘self-worth’’, ‘‘hope’,’ and ‘‘caregivers’ support and acceptance’’» [ (Storick et al, 2019), page 9], could also, or not, be included in palliative care artificial intelligence studies, although health records seldom have that kind of information.

Response: This is a very important point. The key difficulty here will in our opinion be - as you already assume - that data on these aspects is often not collected in a structured way. The lack of data on these important factors probably warrants a commentary on its own. We have however included more use cases where AI could be considered to improve palliative care research:

“While the timing of palliative care is certainly an important aspect that could benefit from AI, it is far from being the only one. Every decision where the clinician has to weigh the possible benefits of an intervention with the stress caused by performing it could benefit from better predictions of whether the benefits that doctor and patient hope for will actually occur. This includes predicting if the pain caused by a bone metastasis will respond to palliative radiation or how long whole brain radiotherapy can delay further neurologic deterioration. One might also try to assess more accurately whether palliative chemotherapy will result in a quicker decline in quality of life than the disease itself for a given patient.”

Thank you for your contributions.

Reviewer 4 Report

This is an interesting commentary about the role of artificial intelligence in palliative care.   This topic is of high relevance to oncology and is important for further exploration as the authors suggested that there is currently a lack of evidence-based research in this field.  The following are areas for further improvement to strengthen the quality of this commentary:

1.  AI is a very broad term that encompasses diverse aspects in computer science with different sub-types.  In the introductory paragraph, the authors need to clearly define what artificial intelligence (AI) means in the health care context, particularly in application to the field of oncology/palliative care.

2.  The commentary emphasizes mainly on the contribution of AI to improving the timing of palliative care (ie. importance of early access to palliative care).  However there is a lack of discussion about specific examples of how AI contributes to enhance the quality of palliative care.  In order to answer this question properly, the authors first need to describe the meaning of "what constitutes high quality palliative care from both the clinician and patient's perspectives?"

3. The concluding statements will benefit from describing the future research implications related to the role of AI in palliative care related to the following priority areas: (1) diagnosis; (2) symptoms management and comfort measures; (3) advanced care planning; (4) end of life/hospice care.

Author Response

Dear reviewer, 

thank you for your in-depth review of our commentary as well as the positive assessment of our work and its purpose. Your suggestions to further strengthen it are highly appreciated and have caused the following changes and additions:

1)  AI is a very broad term that encompasses diverse aspects in computer science with different sub-types.  In the introductory paragraph, the authors need to clearly define what artificial intelligence (AI) means in the health care context, particularly in application to the field of oncology/palliative care.

Response: We have added a definition of AI and why deep learning in particular has gained so much traction in clinical medicine:

“While definitions of what exactly constitutes AI vary, most definitions mention computer-based systems solving tasks that would normally require ‘natural’, especially human, intelligence [3]. Among the many subspecialities of AI, deep learning, where information from the input is extracted through many layers normally by so called neural networks, has emerged as an important tool due to its ability to extract meaningful information from imaging data [4].”

2)  The commentary emphasizes mainly on the contribution of AI to improving the timing of palliative care (ie. importance of early access to palliative care).  However there is a lack of discussion about specific examples of how AI contributes to enhance the quality of palliative care.  In order to answer this question properly, the authors first need to describe the meaning of "what constitutes high quality palliative care from both the clinician and patient's perspectives?"

Response: We have added use cases for AI in palliative care research. Instead of providing a definition of high quality care, we have tried to provide the readers with a general idea of which research questions could benefit: 

“While the timing of palliative care is certainly an important aspect that could benefit from AI, it is far from being the only one. Every decision where the clinician has to weigh the possible benefits of an intervention with the stress caused by performing it could benefit from better predictions of whether the benefits that doctor and patient hope for will actually occur. This includes predicting if the pain caused by a bone metastasis will respond to palliative radiation or how long whole brain radiotherapy can delay further neurologic deterioration. One might also try to assess more accurately whether palliative chemotherapy will result in a quicker decline in quality of life than the disease itself for a given patient.”

3) The concluding statements will benefit from describing the future research implications related to the role of AI in palliative care related to the following priority areas: (1) diagnosis; (2) symptoms management and comfort measures; (3) advanced care planning; (4) end of life/hospice care.

Response: This is an important aspect which made us add a concluding statement that references the aforementioned use cases: 

“As detecting diseases is already a major focus of AI research in medicine and the suggested use cases for leveraging AI in palliative care could impact adequate symptom management as well as advanced care planning and end of life care, one can hope for improvements in all areas of palliative care using the different modalities of AI.” Thank you for your contributions.

Round 2

Reviewer 1 Report

Thanks to the authors; this is an improved commentary.

Author Response

Dear reviewer,

we thank you for the positive assessment of the changes we have made as well as your contributions in the previous review round.

Reviewer 2 Report

Thanks for revising the manuscript based on my comments. After reading the revised version, I understand that there is a lack of research on using AI in palliative care and also the difficulties in implementing AI in their area. However, the feasibility of AI in palliative care research and the importance of AI in that area should be enhanced. As an example to enrich the content, AI is a data-driven method, which relies on data. In palliative care, there is intensive research on various topics including the timing, quality of life, etc, which can provide data to work with at the beginning stage.      

Author Response

Dear reviewer, 

thanks again for your additional contributions and the positive assessment of the changes we have made. We appreciate your suggestions to further strengthen the comment:

1) Thanks for revising the manuscript based on my comments. After reading the revised version, I understand that there is a lack of research on using AI in palliative care and also the difficulties in implementing AI in their area. 

Response: We are confident that these changes based on your suggestions will help the readers understand the purpose of our comment and clarify the underlying chain of arguments. 

2) However, the feasibility of AI in palliative care research and the importance of AI in that area should be enhanced. As an example to enrich the content, AI is a data-driven method, which relies on data. In palliative care, there is intensive research on various topics including the timing, quality of life, etc, which can provide data to work with at the beginning stage.

Response: We agree that the rationale why a foundation of solid non-AI research contributes to making AI more feasible would benefit from additional explanation. Also the way AI research can be introduced to gain importance in the field should be clarified. We have therefore included the following passage:

“However, as palliative care is becoming more and more relevant, the possibilities to apply AI techniques to the research questions of the field increase drastically. Since many AI methods require massive amounts of learning data to reach their full predictive potential, well-curated datasets are a mandatory prerequisite [4]. As ‘conventional’, non-AI research in palliative care becomes increasingly established due to more patients that can be analyzed as well as more funding and interest, those datasets that are created as a byproduct can be used as a starting point for introducing AI to the field. If initial results are promising, a collaboration of palliative care centers to create even larger datasets could further improve those results. Especially imaging studies using deep learning could benefit from more generalizable predictions if the training images are acquired on a variety of  heterogeneous scanners instead of the same machines from a single hospital [4].  To reduce the amount of data needed for the initial studies, researches should also take advantage of advances in AI research such as image augmentation that apply modifications to training images to expand those datasets [13].” 

Thank you for your contributions.

Round 3

Reviewer 2 Report

I don't have further comments on this paper.